

# Supervised ensemble learning methods towards automatically filtering Urdu fake news within social media

Muhammad Pervez Akhter[1], Jiangbin Zheng[1], Farkhanda Afzal[2], Hui Lin[3], Saleem Riaz[3] and Atif Mehmood[4]

[1] School of Software and Microelectronics, Northwestern Polytechnical University, Xian, China
[2] Department of Humanities and Basic Sciences, MCS, National University of Sciences and Technology, Islamabad, Pakistan
[3] School of Automation, Northwestern Polytechnical University, Xian, China
[4] School of Artificial Intelligence, Xidian University, Xian, China

## ABSTRACT

The popularity of the internet, smartphones, and social networks has contributed to the proliferation of misleading information like fake news and fake reviews on news blogs, online newspapers, and e-commerce applications. Fake news has a worldwide impact and potential to change political scenarios, deceive people into increasing product sales, defaming politicians or celebrities, and misguiding visitors to stop visiting a place or country. Therefore, it is vital to find automatic methods to detect fake news online. In several past studies, the focus was the English language, but the resource-poor languages have been completely ignored because of the scarcity of labeled corpus. In this study, we investigate this issue in the Urdu language. Our contribution is threefold. First, we design an annotated corpus of Urdu news articles for the fake news detection tasks. Second, we explore three individual machine learning models to detect fake news. Third, we use five ensemble learning methods to ensemble the base-predictors' predictions to improve the fake news detection system's overall performance. Our experiment results on two Urdu news corpora show the superiority of ensemble models over individual machine learning models. Three performance metrics balanced accuracy, the area under the curve, and mean absolute error used to find that Ensemble Selection and Vote models outperform the other machine learning and ensemble learning models.

## INTRODUCTION

Fake news is also known as deceptive news or misinformation. A news story is a piece of fake news if its authenticity is verifiable false, and it intends to mislead the reader. As compared to fake news, the authenticity of legitimate news is verifiable real, and it plans to convey authentic information to the users (*Abonizio et al., 2020*). Fake news can take on numerous structures including, edited text stories, photoshopped pictures, and unordered video clips. Fake news is similar in appearance to legitimate news, but the aims are different. The aims of spreading fake news are multipurpose, including deceiving readers into benefiting the author, propaganda about a politician to win the election, increased sale of a product by posting fake positive reviews to benefit a businessman, and

Corresponding author
Farkhanda Afzal,
farkhanda@mcs.edu.pk

defame a showbiz star (*Monteiro et al., 2018*). There are numerous hazardous impacts on our society of the proliferation of fake news. Fake news changes the manner of the individual to interpret and reply to legitimate news. Besides, fake news makes individuals skeptical by destroying consumers' trust in the media by posting fabricated and biased news stories (*Agarwal & Dixit, 2020*).

Spreading fake news is not a new problem in our time. Before the advent of the internet, fake news was transmitted through face-to-face (oral), radio, newspaper, and television. In recent years with the advent of the computer, the internet, smartphones, websites, news blogs, and social media applications have contributed to transmitting fake news. There are several reasons for spreading fake news through the internet and social media. It requires less cost and time than traditional news media. It is very easy to manipulate legitimate digital news and share the fabricated news story rapidly. Since 2017, there has been a 13% global increase in social media users (*Kaur, Kumar & Kumaraguru, 2020*). Fake news influences different groups of people, products, companies, politicians, showbiz, news agencies, and businessman.

It requires more energy, cost, and time to manually identify and remove fake news or fake reviews from social media. Some previous studies conclude that humans perform poorly than automated systems to separate legitimate news from fake news (*Monteiro et al., 2018*). For the last few years, machine learning methods' focus is to differentiate between fake and legitimate news automatically. After the U.S. presidential elections in 2015, few popular social media applications like Twitter, Facebook, and Google started to pay attention to design machine learning and natural language processing (NLP) based mechanisms to detect and combat fake news. The remarkable development of supervised machine learning models paved the way for designing expert systems to identify fake news for English, Portuguese (*Monteiro et al., 2018*; *Silva et al., 2020*), Spanish (*Posadas-Durán et al., 2019*), Indonesian (*Al-Ash et al., 2019*), German, Latin, and Slavic languages (*Faustini & Covões, 2020*). A major problem of machine learning models is that different models perform differently on the same corpus. Their performance is sensitive to corpus properties like corpus size, distribution of instances into classes (*Pham et al., 2021*). For example, the performance of K-nearest neighbor (KNN) depends on the number of nearest points ($k$) in the dataset. Support Vector Machine (SVM) suffers from numerical instability when solving optimization problems (*Xiao, 2019*). Similarly, the performance of an artificial neural network is prone to optimal architecture and tuning its parameters (*Pham et al., 2021*).

Ensemble learning is considered an efficient technique that can boost the efficiency of individual machine learning models, also called base-models, base-predictors, or base-learners, by aggregating the predictions of these models in some way (*Lee et al., 2020*). Ensemble learning aims to exploit the diversity of base-predictors to handle multiple types of errors to increase overall performance. Ensemble learning techniques show superior performance in various recent studies about fake news detection. In a recent study, the ensemble learning technique outperformed the four deep learning models including the deep structured semantic model with RNN, intentCapsNet, LSTM model, and capsule neural network (*Hakak et al., 2021*). In another recent study, *Mahabub (2020)* applied

eleven machine learning classifiers including the neural network-based model MLP on a fake news detection corpus. After that, three out of eleven machine models were selected to ensemble a voting model. Ensemble voting with soft voting outperformed the other models. *Gutierrez-Espinoza et al. (2020)* applied two ensemble methods bagging and boosting with SVM and MLP base-predictors to detect fake reviews detection. Experiments show that boosting with MLP outperforms the other.

This can be achieved in numerous ways, including homogenous models with diverse parameters, heterogeneous models, resampling the training corpus, or using different methods to combine predictions of base-predictors (*Gupta & Rani, 2020*). Ensemble learning can be of two types: parallel and sequential. In the parallel ensemble, base-predictors are trained independently in parallel. In the sequential ensemble, base-predictors are trained sequentially, where a model attempts to correct its predecessor (*Pham et al., 2021*). Ensemble learning methods have shown good performance in various applications, including solar irradiance prediction (*Lee et al., 2020*), slope stability analysis (*Pham et al., 2021*), natural language processing (*Sangamnerkar et al., 2020*), malware detection (*Gupta & Rani, 2020*), traffic incident detection (*Xiao, 2019*). In the past, several studies explored machine learning models for fake news detection task in a few languages like Portuguese (*Monteiro et al., 2018*; *Silva et al., 2020*), Spanish (*Posadas-Durán et al., 2019*; *Abonizio et al., 2020*), Urdu (*Amjad et al., 2020*; *Amjad, Sidorov & Zhila, 2020*), Arabic (*Alkhair et al., 2019*), Slavic (*Faustini & Covões, 2020*; *Kapusta & Obonya, 2020*), and English (*Kaur, Kumar & Kumaraguru, 2020*; *Ozbay & Alatas, 2020*). As compared to machine learning, a few efforts have been made to explore ensemble learning for fake news detection like Indonesian (*Al-Ash & Wibowo, 2018*; *Al-Ash et al., 2019*), English (*Kaur, Kumar & Kumaraguru, 2020*; *Sangamnerkar et al., 2020*). Therefore, this study aims to investigate ensemble learning methods for the fake news detection task.

Urdu is the national language of Pakistan and the 8th most spoken language globally, with more than 100 million speakers (*Akhter et al., 2020a*). Urdu is the South Asian severely resource-poor language. As compared to resource-rich languages like English, a few annotated corpus from very few domains are available for research purposes. Besides, insufficient linguistic resources like stemmers and annotated corpora make the research more challenging and inspired. Particularly in Urdu, studying fake news detection has several challenges. First, unavailability of some sufficient annotated corpus. A recent study (*Amjad et al., 2020*) proposed an annotated fake news corpus with a few hundred news articles. Experiments on this corpus reveal the poor performance of machine learning models. Second, labeling a news article as "fake" or "legitimate" needs experts' opinions, which is time taking. Last, hiring experts in the relevant domains is an expensive task. Therefore, in this study, we design a machine-translated corpus of Urdu news articles translated from English news articles using Google Translate. We followed the same procedure in the study (*Amjad, Sidorov & Zhila, 2020*). Experiments reveal that machine learning models do not perform well on machine-translated corpus compared to the real dataset (*Amjad, Sidorov & Zhila, 2020*). Because of the small size, the corpus is not sufficient to make any conclusion about machine learning models' performance.

Further, to the best of our knowledge, no study explores ensemble learning models for Urdu fake news detection tasks.

Inspired by the work done in other languages, we are investigating the issue of fake news detection in the Urdu language. The major aim of this study is to explore the capability of ensemble learning models in improving fake news predictions in resource-poor language Urdu. Our significant contributions to this study have been summarized below:

- We manually built an annotated news corpus composed of Urdu news articles distributed into legitimate and fake categories.
- We perform several experiments using three diverse traditional machine learning classifiers Naïve Bayes (NB), Decision Tree (DT), and SVM, and five ensemble models, including Stacking, Voting, Grading, Cascade Generalization, and Ensemble Selection, to achieve improved prediction quality relative to conventional individual machine learning models.
- We investigate the performance of our models using three feature sets generated through character-level, word-level, and statistical-based feature selection methods.
- We report experiments of both machine learning and ensemble learning models on two fake news corpora of the Urdu language.
- We comparatively analyze the performance of our models using four performance measures, including balanced accuracy, the area under the curve, time and mean absolute error.

Hence forward this article is organized as follows: "Related Work" presents the essential related works. "Machine Learning and Ensemble Learning Models" provides a brief overview of machine learning and ensemble learning models used in this study. "Methodology and Corpus Construction" will show the architecture of the adopted framework and corpus characteristics. The results of the experiments are comparatively discussed in "Results". Finally, "Conclusions" ends the article with conclusions and future directions.

## RELATED WORK

Online social media and instant messaging applications like Facebook, Google, and Twitter are popular these days in talking to your loved ones, expressing your opinion, sharing professional information, or posting news about the subject of interest. Further, it is common to find some information on the internet quickly. Unfortunately, all the information available on social media sites is not accurate and reliable as it is straightforward to manipulate digital information and quickly spread it in the world. Therefore, it is vital to design some accurate, efficient, and reliable automated systems to detect fake news from a large corpus.

In the past, numerous machine learning methods have been used to combat fake news. *Monteiro et al. (2018)* showed that the multi-layer perceptron (MLP) model outperforms the NB and random forest to identify fake news from a large news corpus.

The study of *Faustini & Covões (2020)* concludes that SVM with bag-of-word (BoW) feature outperformed the other on five corpora of three languages Germanic, Latin, and Slavic. A benchmarking study for fake news detection concludes that SVM with linguistic-based word embedding features enables us to classify fake news with high accuracy (*Gravanis et al., 2019*). A study about Portuguese fake news detection reveals that random forest outperforms the other five machine learning models (*Silva et al., 2020*). AdaBoost achieves the best performance on a small corpus than the other six models to separate fake news from legitimate news (*Amjad et al., 2020*). A recent study of fake news detection using supervised artificial intelligence methods shows that the DT is the best model out of twenty-three models (*Ozbay & Alatas, 2020*). After analyzing the above studies, we can conclude designing an effective and high-performing system with a careful selection of the machine learning model and the feature selection method.

To overcome individual machine learning models' issues and increase the classification performance, an ensemble of several diverse machine learning models has shown superior performance than individual machine learning in several applications. *Xiao (2019)* applied ensemble techniques with SVM and KNN base learners to detect traffic incidents. Experiments show the superiority of the ensemble model over individual machine learning models. A recent study about detecting fraud in credit cards concludes that the ensemble approach based on cost-sensitive meta-learning outperforms traditional machine learning models. *Pham et al. (2021)* used four ensemble models: boosted trees, bagged trees, random forest, and generalized random forest for slope stability analysis. Experimental results prove the best performance of extreme gradient boosting over other ensemble models and machine learning models. For malware detection (*Gupta & Rani, 2020*) used five base-predictors, and the output of each base-predictor was ranked by calculating and aggregating the output weights. Then using two ensemble techniques Voting and Stacking to rank the output. A higher value ranked by the ensemble technique was the final prediction of the model. After extensive experiments, the study reveals that weighted Voting shows the best performance than Stacking.

Compared to other applications, fake news detection using ensemble learning techniques has very few studies in the past. *Kaur, Kumar & Kumaraguru (2020)* proposed a multi-level Voting model for the fake news detection task. The study concludes that the proposed model outperforms the other 11 individual machine learning and ensemble learning models. For multiclass fake news detection *Kaliyar, Goswami & Narang (2019)* and *Abonizio et al. (2020)* used Gradient Boosting ensemble techniques and compare their performance with several individual machine learning models. Results on multiple corpora show that Gradient Boosting achieves the best performance than any other individual models. A recent study (*Huang & Chen, 2020*) proposed a self-adaptive harmony search algorithm to get optimized weights of ensemble models. The proposed algorithm achieved outstanding performance with 99.4% accuracy. The Bagging approach to detect fake news showed superior performance than SVM, Multinomial Naïve Bayes, and Random Forest (*Al-Ash et al., 2019*).

English is a resource-rich language, and many linguistic resources are publically available for research purposes. Therefore, several research studies have been performed

**Table 1 Resource-poor language corpora for fake news detection.**

| Corpus | Language | Legitimate | Fake | References |
|---|---|---|---|---|
| – | Chinese | 131 | 187 | *Zhang et al. (2009)* |
| Slovak National Corpus | Slovak | 80 | 80 | *Kapusta & Obonya (2020)* |
| DECOUR | Italian | 1202 | 945 | *Fornaciari & Poesio (2013)* |
| – | English and Spanish | 100 | 100 | *Pérez-Rosas et al. (2018)* |
| CSI | Dutch | 270 | 270 | *Verhoeven & Daelemans (2014)* |
| Fake.Br | Portuguese | 3,600 | 3,600 | *Silva et al. (2020)* |
| TwitterBR | Brazilian | 4,589 | 4,392 | *Faustini & Covoes (2019)* |
| Btvlifestyle | Bulgarian | 69 | 69 | *Hardalov, Koychev & Nakov (2016)* |
| – | Bangla | 1,548 | 993 | *Hussain et al. (2020)* |
| Spanish Fake News | Spanish | 491 | 480 | *Posadas-Durán et al. (2019)* |
| Bend the Truth | Urdu | 500 | 400 | *Amjad et al. (2020)* |
| MT | Urdu | 400 | 400 | *Amjad, Sidorov & Zhila (2020)* |
| UFN | Urdu | 1,032 | 968 | Our corpus |

for the fake news detection task. A study gives a comparison of 23 publically available datasets of English (*Sharma et al., 2019*). A recent survey compares different techniques to identify fake news, their credibility detection, and discusses fundamental theories and opportunities (*Zhou & Zafarani, 2020*). There is a severe lack of research studies for fake news detection in languages other than English. For the Indonesian language, a study by *Al-Ash et al. (2019)* shows that the bagging model outperforms the three individual models: SVM, Random Forest, and Multinomial Naïve Bayes. In *Abonizio et al. (2020)* applied three machine learning models (KNN, SVM, and random Forest) and extreme gradient boosting (ensemble model) on five datasets of three languages (English, Portuguese, and Spanish). In another study about fake news detection for Portuguese, random forest shows high accuracy in most of the experience (*Silva et al., 2020*). Extreme gradient boosting shows the best performance than other individual models. For the Urdu language (*Amjad et al., 2020*), Adaboost outperforms the other seven machine learning models on a very small corpus. DT improves the classification accuracy for fake news detection for the Slovak language (*Kapusta & Obonya, 2020*).

The lack of availability of a benchmarked annotated corpora of resource-poor languages are the major problem to investigate and compare the performance of numerous automated methods for fake news detection. Therefore, in several other than English studies, authors designed their corpus by collecting news articles from the internet and other web resources and manually annotating these articles into fake and legitimate news. A list of corpora for several resource-poor languages is given in Table 1. It can be noticed that all the corpora except "Fake.Br" are very small in size. Because corpus designing is a costly and time-consuming task, the annotation process requires several experts from various fields to decide about the news article (*Amjad, Sidorov & Zhila, 2020*). To date, our proposed corpus Urdu fake news (UFN) is the largest corpus than others.

# MACHINE LEARNING AND ENSEMBLE LEARNING MODELS

## Machine learning models

This section gives a brief overview of three traditional machine learning models: Naïve Bayes, Decision Tree, and Support Vector Machine. We also described their significant drawbacks, which limit their performance on various tasks.

### Naïve Bayes

Naïve Bayes used a probabilistic approach based on Bayesian Theorem with two assumptions: (1) all the features are independent of each other in the dataset, and (2) all the features have equal effects. It is simple, popular, and useful for the classification of a large corpus, although the corpus does not hold independence. NB is challenging to interpret for a large corpus, and its assumption about features independence makes its performance poor, especially when the data distribution is very skewed (*Komiya et al., 2011*). Several studies have used NB for fake news detection tasks like for Portuguese (*Monteiro et al., 2018*) and English (*Gravanis et al., 2019*).

### Decision tree

The decision tree algorithm learns a decision rule inferred from the training data to design a decision tree model. Nodes of the tree represent the feature vectors taken from the text of the news article. Leaf nodes represent the set of possible labels or classes in the corpus. In our case, there are two possible labels: fake and legitimate. The DT predicts the article's label by learning features from the tree's root to one of the leaf nodes. It is simple and easy to interpret because all the information about the model behavior and influential variables is available. Therefore, the DT is also known as a white-box model. Drawbacks of the DT include overfitting and instability, a complex tree for a high-dimensional dataset that is not easy to interpret (*Pham et al., 2021*). For the fake news detection task, DT has shown good performance for Slovak (*Kapusta & Obonya, 2020*), Portuguese (*Silva et al., 2020*), English (*Gravanis et al., 2019*; *Ozbay & Alatas, 2020*), and Urdu (*Amjad et al., 2020*) languages.

### Support vector machine

Support vector machine is a non-parametric machine learning model. The performance of SVM is usually considered suitable for binary classification tasks with high-dimensional data. SVM maps all the features obtained from news articles' text into n-dimensional space where a feature represents the particular coordinate. During training, SVM learns a hyper-plan that best discriminates the features of one class to another. Therefore, SVM is also known as a discriminative classifier. Although SVM performs well with high-dimensionality data, it is difficult to interpret, requires significant computing resources, and faces numerical instability for optimization problems (*Pham et al., 2021*). SVM shows excellent performance for fake news detection task in several studies of various languages like English (*Monteiro et al., 2018*; *Gravanis et al., 2019*), Urdu (*Amjad, Sidorov & Zhila, 2020*), Portuguese (*Silva et al., 2020*), Dutch (*Verhoeven & Daelemans, 2014*), Germanic, Latin, and Slavic (*Faustini & Covões, 2020*).

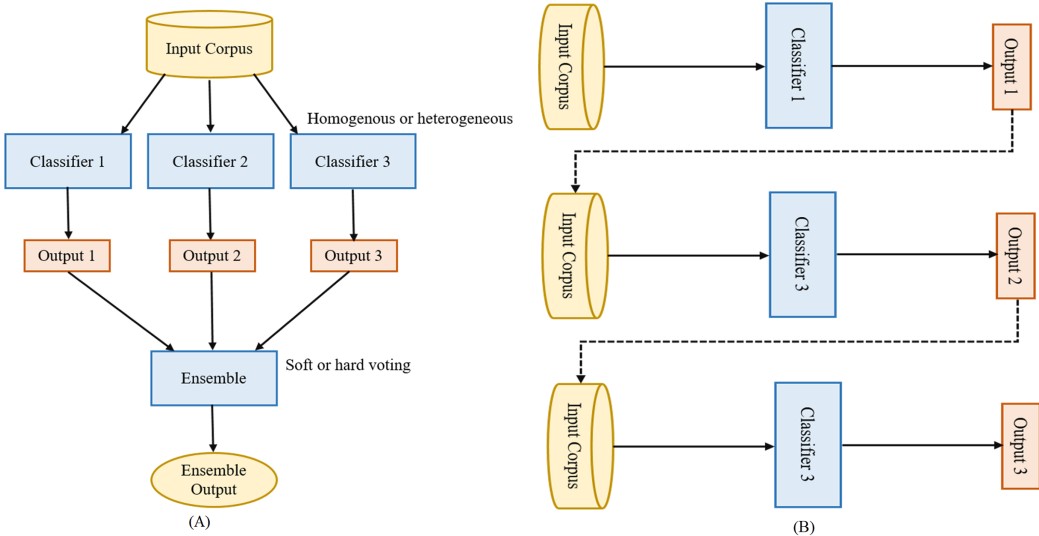

**Figure 1 Parallel vs. sequential ensemble.** (A) Parallel and (B) sequential ensemble.

## Ensemble learning models

Ensemble learning aggregates the individual machine learning models (base-predictors) to design a superior model to increase overall accuracy by handling the shortcomings of the base-predictors. It is known as the most efficient technique for improving the performance of machine learning models. Nowadays, ensemble learning methods are gaining more popularity than traditional individual machine learning models in numerous classification tasks like fake news detection (*Kaur, Kumar & Kumaraguru, 2020*), malware detection (*Gupta & Rani, 2020*). Ensemble learning methods fall into two categories: parallel ensemble and sequential ensemble. Both are shown in Figs. 1A and 1B. In the parallel ensemble, the base-predictors are trained in parallel on the input corpus. The parallel ensemble has the advantages of simultaneous predictions, utilizing different CPU cores to execute the models simultaneously, and utilizing the characteristics of independence among them. In the sequential ensemble, the base-predictors are trained sequentially where the output of the one base-predictor plus the input corpus is the input to the next base-predictor. In other words, the base-predictors are dependent on each other. The next base-predictor challenge is to try to correct the errors of the previous base-predictor to improve the overall prediction accuracy (*Pham et al., 2021*). Base-predictors can be homogenous or heterogeneous. In homogenous, a single machine learning model (like DT or NB) is trained in parallel or sequentially, while in heterogeneous different machine learning models (like DT and NB) are trained in parallel or sequentially. The ensemble learning method is advantageous if the heterogeneous machine learning models are used as base-predictor (*Kittler, Hater & Duin, 1996*). Heterogeneous ensemble learning can be performed using different classifiers with different feature sets, training sets, and evaluation methods. In this section, we provide a brief description of the five ensemble models used in this study.

### Stacking

Stacking model ensembles several base-predictors machine learning models using the stacking method. It was initially proposed by *Ting & Witten (1997)* and used in several studies for classification tasks like malware detection (*Gupta & Rani, 2020*), credit card fraud detection (*Olowookere & Adewale, 2020*), and spam detection (*Saeed, Rady & Gharib, 2019*). It can perform both classification and regression on data. Base-predictors are trained on input data, and the output of these base-predictors is given to a meta-classifier which makes the final prediction about the class of an input sample. Meta-classifier can be any classifier like Adaboost, Regression, etc. The primary aim of meta-classifier is to learn the optimal weights to combine the predictions of base-predictors and produce better prediction results than individual base-predictor results. Therefore, Stacking reduces the variance and improve the quality of classification. For unseen test articles from the test set, the article is passed to the base-predictor to classify these test articles. Their classification is then passed to stacking-based ensemble learners to make the final classification of that article as either fake or legitimate.

### Voting

Voting is a meta-classifier that combines several base-predictors using different combination rules. Base-predictor can be any machine learning model. Individual base-predictors are trained on training data. The output of these base-predictors is combined using some combination rules like majority voting, minimum or maximum probabilities, or product of probabilities (*Kittler, Hater & Duin, 1996*). Majority voting is also known as hard-voting as the class with majority votes is considered the input article's final class. In soft-voting, the final class is a class with the highest probability averaged over the individual predictors (*González et al., 2020*). Voting method have used in several classification tasks like fake news detection (*Kaur, Kumar & Kumaraguru, 2020*), spam detection (*Saeed, Rady & Gharib, 2019*), and slope stability analysis (*Pham et al., 2021*).

### Grading

Grading is an ensemble model originally presented by *Seewald & Fürnkranz (2001)*. Grading is a type of meta-classifier that corrects the graded predictions of base-predictors at the base-level assuming that different base-predictors make different errors. Graded predictions are those predictions that are marked as incorrect or correct predictions by the base-predictor. A meta-classifier is learned for each base-predictor whose basic job is to predict when the base-predictor will error. These meta-classifiers are trained on a training set constructed from the graded predictions of corresponding base-predictors as new class labels. Grading is different from Stacking that uses the incorrect predictions of base-predictors as the attribute value for meta-classifier. Several studies show that Grading outperforms the Voting and Stacking ensemble models on classification tasks (*Seewald & Fürnkranz, 2001*; *González et al., 2020*).

### Cascade generalization

Cascade Generalization belongs to a special Stacking generalization that uses a sequentially layered architecture for combining multiple classifiers. The predictions of several

base-predictors are used in the next stage for final prediction (*Gama & Brazdil, 2000*). An extension of the original data is achieved at each level by inserting new attributes derived from the probability class distribution given by the base-predictors. Cascade Generalization is different from Stacking generalization in that Stacking is parallel, while Cascade is sequential in nature. Because of its sequential nature, intermediate models have access to the original attributes and the low-level models' predictions. Cascade provides the rules to choose the high-level and low-level models for classification. A major problem of the Cascade is that it transforms the feature space into a new high-dimensional feature space (the curse of dimensionality) that sometimes leads to a more difficult learning problem (*Gama & Brazdil, 2000*).

### Ensemble selection

Ensemble selection is a method to construct an ensemble of several machine learning models. It starts with an empty ensemble and adds a model into the ensemble if it increases the performance of the ensemble. This process is repeated up to a specified number of iterations or until all the models have been used (*Caruana, Ksikes & Crew, 2014*). Models are added into an ensemble using numerous ways like forwarding selection, backward elimination, and the best model. It stops adding models into the ensemble when the ensemble's performance starts to decrease after achieving the best performance. Ensemble selection allows ensembles to be optimized to performance metrics such as accuracy, cross-entropy, mean precision, or ROC Area (*Ballard & Wang, 2016*; *Nguyen et al., 2020*). In a recent study, Ensemble Selection outperforms the other ensemble models to classify 62 datasets (*Nguyen et al., 2020*).

## METHODOLOGY AND CORPUS CONSTRUCTION

The proposed framework for fake news detection consists of four-phases. The first phase describes the procedure to design a corpus of Urdu news articles. The second phase explains the preprocess operations performed on the text of news articles. The third phase shows feature selection or dimensionality reduction approaches. The fourth phase provides the description of individual machine learning models or base-predictors for ensemble models. At last, the fifth phase describes the numerous ensemble learning models used in this study. The architecture with five layers is shown in Fig. 2.

### Corpus design

In this study, we choose two corpora of text news articles of Urdu language for experiments. As Urdu is a resource-poor language, there is no standard corpus available for fake news detection task to the best of our knowledge. Because of the lack of linguistic resources, the collection of news articles from multiple sources is a tough task. Besides, the annotation process of these news articles based on the articles contents needs expert knowledge, a lot of time, and budget. Therefore, augmented corpus design is the only way to perform research about fake news detection for resource-poor languages. Our first corpus is UFN augmented corpus. It contains two thousand news articles randomly selected and translated from English language fake news corpus using online Google Translator. The original English corpus contains nearly 8,000 news articles. We picked a

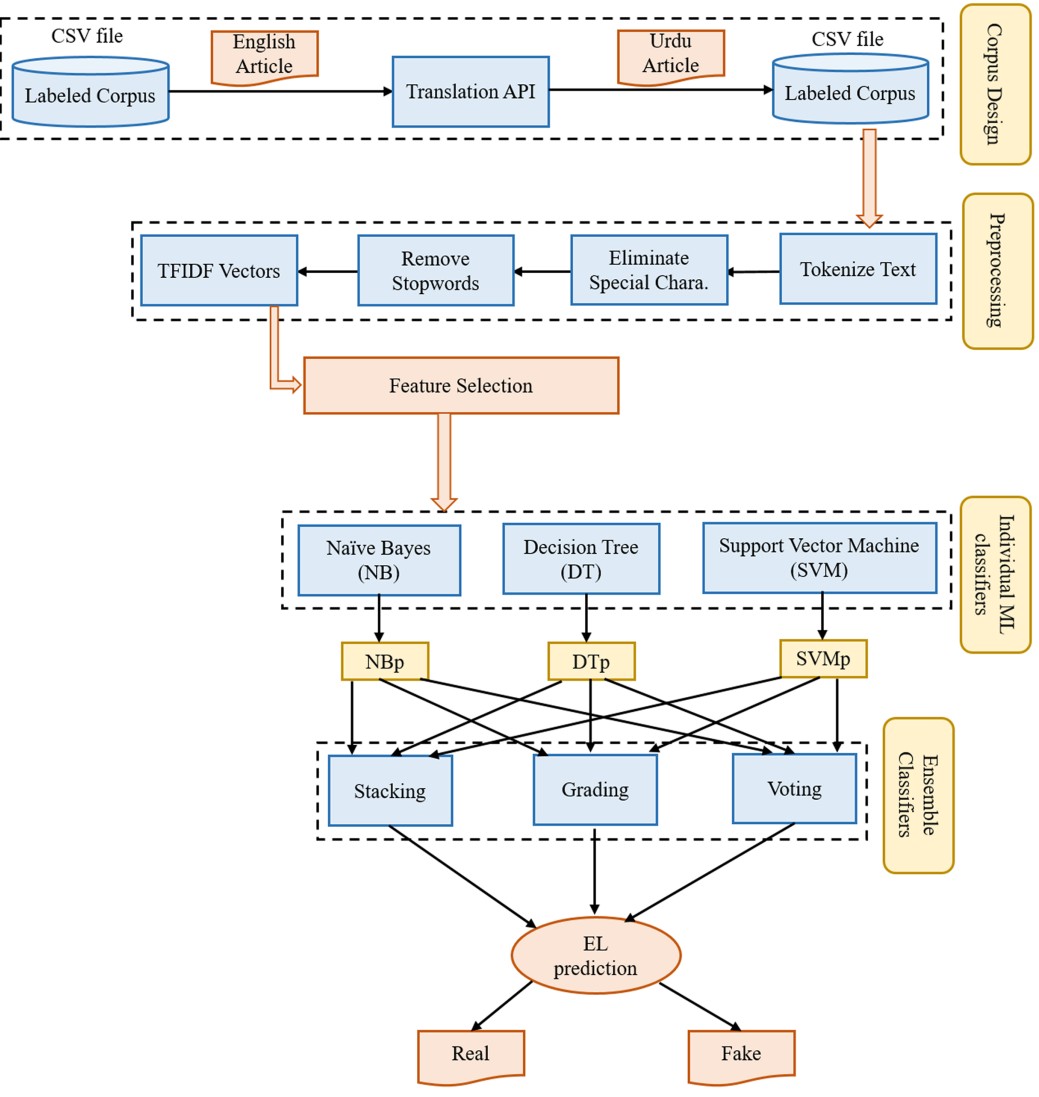

**Figure 2 The architecture of the proposed framework used for fake news detection.**

subset of two thousand articles because (1) manual translation of all the articles in the original corpus is time taking and difficult to perform, (2) English-Urdu translation using Google Translator is not hundred percent accurate and to the best of our knowledge, no study investigates this problem. This is an open research question and can be investigated in future studies, (3) we aim to explore the detection performance of ensemble learning techniques. Several recent studies about fake news detection in Slovak (*Kapusta & Obonya, 2020*), Italian (*Fornaciari & Poesio, 2013*), and Spanish (*Posadas-Durán et al., 2019*) used corpora with even less than two thousand news articles (see Table 1).

Our second corpus is a small size Bend the Truth (BET) corpus designed and annotated by *Amjad et al. (2020)*. This corpus contains only 900 original news articles in Urdu. A sample of the Urdu news articles is shown in Table 2. After translation, the Urdu article label was the same as in the English corpus's corresponding article. The final corpus is

**Table 2  Samples of the legitimate and fake news.**

| Legitimate | Fake |
|---|---|
| ، امریکہ عالمی جنگ 3 کی تیاری کر رہا ہے - 31 اکتوبر بجے شام 4:52 بجے انتخابی خدمت میں شامل 2016 خواتین کے علاوہ خوبصورت حقیقت پسندی۔ امریکی فوج ابھی بھی صرف رضاکارانہ ہے اور ویتنام کی جنگ کے بعد سے اس کا مسودہ نہیں رہا ہے۔ خواتین کے مسودے کے پارے میں فوج کے ایک 4 اسٹار جنرل نے یہ تبصرہ کیا تھا اور انہوں نے کہا تھا کہ لبرل یابو کو بند کردیں۔ | اجمہوری امیدواروں کے لئے مباحثے کے ابداف - لاس ویگاس میں 2016 کی دوڑ کے پہلے جمہوری صدارتی مباحثے کے دوران منگل کو پانچ امیدوار اسٹیج پر ہوں گے۔ شام کے اختتام پر بہتر پوزیشن میں دور آنے کے لئے ہر ایک کو کیا کرنا ہے یہ یہاں ہے: |
| اصلی انکشاف! سیکریٹ ایلین بیس مون کے ٹائکو کرٹر میں ملا - اصلی انکشاف! چاند کے ٹائکو کریکٹر میں ملنے والا خفیہ ایلین بیس # گرے 52 اصلی انکشاف وہ جگہ ہے جہاں آپ کو قمری سطح پر کچھ مل جاتا ہے جو ممکنہ طور پر موجود نہیں ہوسکتا جب تک کہ کوئی اسے نہ بنائے۔ کوئی راستہ نہیں یہ قدرتی تشکیل ہے --- کچھ زاویہ اجنبی les  ایسی تعمیر کی گئی ہے جو ---- 90 انسان ساختہ بابمی تعامل کے بغیر ممکن ہی نہیں ہے۔ / مزید 'سگریٹ نوشی بندوق' بیرون ملک سے ذہانت کا ناقابل | انیو یارک ڈیموکریٹک پرائمری محسوس کرنے کی کی پانچ وجوہات - نیو یارک ، نیو یارک؟ € ''اگر ہلیری کلنٹن جمہوری صدارتی دوڑ جیت رہی ہیں تو اسے ایسا کیوں محسوس ہوا جیسے وہ بار رہی ہیں؟ ہاں ، مسز کلنٹن نے منگل کو نیویارک میں ایک اہم فتح حاصل کی ، اور انہوں نے ابتدائی طور پر اپنایا ہوا آبائی ریاست جیت لیا۔ لیکن کیا اس کا نتیجہ کبھی بھی شک میں رہنا چاہئے تھا؟ نیویارک کی سابقہ سینیٹر کی حیثیت سے وہ |

**Table 3  The statistical description of the Urdu language corpora for fake news detection.**

| Corpus | Bend the Truth | | Urdu Fake News | |
|---|---|---|---|---|
| Labels | Fake | Legitimate | Fake | Legitimate |
| Vocabulary | 1,84,023 | 1,20,394 | 11,47,547 | 9,54,254 |
| Minimum Length Article | 57 | 59 | 25 | 25 |
| Maximum Length Article | 2,159 | 1,153 | 7,045 | 6,068 |
| Total Article | 400 | 500 | 968 | 1,032 |

available online on GitHub in CSV file format. The statistics of both corpora are shown in Table 3. It can be noticed that our designed corpus UFN is larger than the BET corpus based on the total number of articles, size of vocabulary, and the length of the article.

## Corpus preparation and preprocessing

Articles in the corpus are in an unstructured format and cannot be processed directly by the machine learning models. We must have to perform a series of operations on the corpus to convert an unstructured corpus into a structured corpus. We have cleaned and processed both corpora's news articles before generating the feature vectors for feature selection. We tokenized the text using space characters. Special characters, email addresses, and website URLs were removed from the text. After cleaning the text, we removed the most frequent and rare words of the Urdu language (also known as stopwords) from the text. The cleaned and the preprocessed articles were converted into numeric feature vectors using the term frequency-inverse document frequency (TF-IDF) method as used in a recent study (*Ozbay & Alatas, 2020*). Both corpora were passed through the same number of preprocessing steps.

## Feature selection

In our experiments, we have performed the experiments using three feature selection methods character tri-grams, BOW, and information gain (IG). A recent study shows the superiority of the character n-gram method over word-level n-grams in short text classification tasks (i.e., offensive language detection) in Urdu text comments (*Akhter et al., 2020b*). Character n-gram is a contiguous sequence of characters in the text. In character n-grams, the value of n is taken as three, which means the combination of three characters makes a tri-gram feature. From the UFN corpus, 1,084 character n-grams, and from the BET corpus 1,091 n-grams were extracted. BOW is a content-based feature representation in which a news article is represented as a set of words that occur in it at least once. IG measures the goodness of the features in the text. A comparative study concludes that IG is the best feature selection method for document-level text classification of Urdu. In our experiments, we have selected the top one thousand IG features from both corpora. A total of 1,225 and 1,214 BoW features from BET and UFN, respectively.

## Heterogeneous machine learning models

In our experiments, for machine learning classification, we use three individual machine learning models NB, SVM, and DT to detect fake news. All three models are heterogeneous. The working of these models is entirely different from each other. Using character-level n-grams from text articles, these models analyze the article's text and classify it into one of the categories legitimate or fake. Detail description of these machine learning models is given in "Machine Learning Models".

## Ensemble learning models

Ensemble classification is usually based on two levels: base-level and ensemble-level. We use three diverse machine learning models, SVM, DT, and NB, as base-predictors at the base-level. Input to these base-predictors is the character-level n-grams extracted from the news articles. Output predictions of these base-predictors are input to ensemble-level models. The basic aim of using the ensemble model is to overcome the base-predictors' shortcomings and improve overall prediction accuracy. We use five ensemble models for ensemble classification, known as Voting, Grading, Stacking, Cascading Generalization, and Ensemble Selection. A brief description of our ensemble models is given in "Naïve Bayes".

## Performance measures

To compare the performance of individual machine learning models and ensemble learning models, in this study, we employed the three well-known performance measures mean absolute error (MAE), balanced accuracy (BA), and area under the curve (AUC).

### Balanced accuracy

Separation of fake news from legitimate news is a binary classification task. A model has to decide about an article, either a legitimate article or a fake article. As the focus of this study is to detect both classes correctly, we used the balanced accuracy performance

**Table 4 Rules for classifying the discrimination using AUC.**

| AUC values | Classifier categories |
|---|---|
| $AUC < 0.5$ | No Discrimination |
| $0.7 \leq AUC < 0.8$ | Acceptable |
| $0.8 \leq AUC < 0.9$ | Excellent |
| $0.9 \leq AUC$ | Outstanding |

measure to compare the performance of our models. Balanced accuracy calculates the average of the proportion of corrects of each class individually. Balanced accuracy can be calculated as follows:

$$\text{Balanced Accuracy (BA)} = \left[\frac{\text{TP}}{\text{TP} + \text{FP}} + \frac{\text{TN}}{\text{TN} + \text{FN}}\right] \bigg/ 2 \tag{1}$$

***Area under the receiver operating characteristic curve***

Area Under the receiver operating characteristic curve, also known as AUC, is used to estimate the performance of a machine learning model using a single value. AUC provides the probability that the model will rank a randomly chosen positive sample higher than a randomly chosen negative sample. AUC can be calculated by Eq. (2). $\text{TP}_{\text{rate}}$ is the ratio of correctly predicted articles as fake articles. It is also known as recall and can be calculated as above in Eq. (3). $\text{FP}_{\text{rate}}$ is the number of legitimate news articles that are misclassified or incorrectly predicted as fake news articles.

$$\text{AUC} = \frac{1 + \text{TP}_{\text{rate}} - \text{FP}_{\text{rate}}}{2} \tag{2}$$

$$\text{FP}_{\text{rate}} = \frac{\text{FP}}{\text{FP} + \text{TN}} \tag{3}$$

General rules for categorizing the performance of the machine learning model using AUC are given in Table 4. These rules are used and discussed in *Pham et al. (2021)*.

***Mean absolute error***

The error refers to the absolute difference between the actual values and the predicted values. MAE measures the average magnitude of the error from a set of predictions made by the model. MAE can be calculated as follows:

$$\text{Mean Absolute Error (MAE)} = \frac{1}{N} \sum_{j=1}^{n} |y_j - \hat{y}_j| \tag{4}$$

# RESULTS

## Experiment setup

As mentioned earlier, in this study, three diverse machine learning models NB, DT, and SVM have been used to classify news articles into legitimate and fake classes. We use a

well-known data mining tool, WEKA, for experiments. WEKA provides a list of supervised and unsupervised machine learning models, data preprocessing techniques, and various performance evaluation methods. Machine learning models have few parameters, called hyper-parameters, to minimize the difference between training error and testing error. In our experiments, we do not fine-tune the hyper-parameters of all these models. We use the default parameters given in the WEKA as the default parameters have already the best values in most of the cases. We use the J48 algorithm for DT implementation. LibLINEAR algorithm for SVM implementation. We use the same DT, SVM, and NB models as base-predictors for all the ensemble models. For Voting and Stacking, along with the three base-predictors, we use Adaboost as a meta-classifier.

## Model training and testing

For training and validation of individual machine learning models and ensemble models, we use k-fold cross-validation as mentioned in "Corpus Design" that both corpora have not been divided into a training subset and testing subset. k-fold cross-validation is a popular choice and used in many past research studies. In our experiments, we use 10-fold cross-validation, where k-1 folds are used for training, and one-fold is used to test the model's prediction performance. This process is repeated ten times to achieve the final performance score.

## Results and discussion of machine learning models

The experiment results achieved using 10-fold cross-validation from individual machine learning models are shown in Table 5. We compare the performance using BA, AUC, MAE, and time. A close observation of the results reveals that SVM outperforms the other for all corpora performance metrics. A model is considered an accurate classifier if its balanced accuracy is higher than the other models. The BA metric shows that SVM outperforms the others on the UFN corpus. SVM achieves BA scores 81.6%, 86.7%, and 87.3% using tri-gram, BoW, and IG features, respectively. SVM also outperforms the others on the BET corpus. It achieves 76.3, 62.7, and 62.4 using tri-gram, BoW, and IG features, respectively. IG feature outperforms the others and achieves the maximum BA score 87.3% on large UFN corpus while tri-gram approach achieves maximum BA scores of 76.3% on BET corpus. With the lowest balanced accuracy scores, NB shows the worst performance. It is noticed that SVM has higher accuracy at UFN than BET. The size of the UFN corpus, in terms of the number of articles and vocabulary size, is almost double the of BET, and SVM is considered a good model for the classification of high-dimensional feature space (*Faustini & Covões, 2020*).

Similarly, AUC scores of the SVM model are the highest score than DT and NB on both corpora. SVM achieves 87.3% and 76.3% AUC metric values on UFN and BET corpora, respectively. Here, again IG proves the best feature selection method for UFN while tri-gram on BET corpus as SVM achieves the maximum AUC scores on IG and tri-gram features. Further, as per the rules of Table 4, a comparison of AUC scores of all the models concludes that the performance of SVM on UFN is excellent ($0.8 \leq AUC < 0.9$)

**Table 5 The evaluation metrics values of supervised individual machine learning models.**

| Ensemble models | BA | | AUC | | MAE | | Time | |
|---|---|---|---|---|---|---|---|---|
| | UFN | BET | UFN | BET | UFN | BET | UFN | BET |
| Naïve Bayes | 64.8 | 67.4 | 69.4 | 75.5 | 35.5 | 34.2 | 0.94 | 0.32 |
| Support Vector Machine | 81.6 | 76.3 | 81.5 | 76.3 | 18.5 | 23.5 | 2.56 | 0.67 |
| Decision Tree | 71.2 | 76.1 | 69.8 | 73.1 | 29.2 | 23.5 | 8.02 | 2.19 |
| BoW | | | | | | | | |
| Naïve Bayes | 75.3 | 56.3 | 80.1 | 61.9 | 24.6 | 43.4 | 0.73 | 0.41 |
| Support Vector Machine | 86.7 | 62.7 | 86.7 | 62.7 | 13.3 | 37.1 | 0.15 | 0.8 |
| Decision Tree | 76.6 | 55.6 | 76.6 | 58.4 | 23.8 | 43.5 | 10.96 | 4.39 |
| IG | | | | | | | | |
| Naïve Bayes | 75.3 | 56.7 | 80.5 | 63.5 | 24.4 | 42.2 | 0.59 | 0.24 |
| Support Vector Machine | 87.3 | 62.4 | 87.3 | 62.4 | 12.7 | 37.7 | 0.19 | 0.17 |
| Decision Tree | 76.8 | 57.0 | 76.8 | 61.5 | 23.6 | 41.6 | 10.28 | 3.56 |

on all the features. On the BET corpus, SVM performance is only acceptable ($0.7 \leq AUC < 0.8$) on tri-gram features. The performance of DT and NB is just acceptable.

From Table 5, it can be seen that in terms of MAE, the prediction error of SVM is the lowest than others. SVM achieves the lowest MAE score 12.7% with IG on UFN while 23.5% with tri-gram on BET corpus. The highest MAE values of NB proves its worst performance to detect Urdu fake news. A model is considered efficient if it takes minimum time than other models to build the model on some corpus. Again, SVM takes a minimum time of 0.15 on BOW and 0.17 s on IG to build the model on UFN and BET. DT takes the highest time on all features to build the model for both corpora. Further, it is notable that all the models perform betters on our designed machine-translated UFN corpus than the original news article's corpus BET. It shows that Google API translated text, preprocessing methods, and feature selection methods all together improve the classification accuracy of our models to detect fake news. Therefore, after analyzing the results, we conclude that SVM is a more accurate, reliable, efficient, and robust classification model among the three models to detect Urdu text's fake news articles.

## Results and discussion of ensemble models

The values of four evaluation metrics balanced accuracy, the area under curve, mean absolute error, and time achieved by five ensemble models on both corpora are given in Table 6. Time and MEA are the two metrics whose minimum values are required by the models. The other two metrics balanced accuracy and AUC, whose maximum values are required to achieve by a model. For the time metric, it is visible that Voting takes the minimum time than other models to build the model on the input corpus. Voting takes 11.52 s and 3.12 s to build the model on UFN and BET corpora, respectively. As the size of the BET model is very small, the Voting takes the minimum time to build the model than all the other models. It can also be noticed that the minimum time taken by

**Table 6 Evaluation metrics for supervised ensemble models using both corpora.**

| Ensemble models | BA | | AUC | | MAE | | Time | |
|---|---|---|---|---|---|---|---|---|
| | UFN | BET | UFN | BET | UFN | BET | UFN | BET |
| Ensemble Selection | 77.8 | 83.3 | 85.9 | 91.0 | 32.2 | 24.0 | 40.13 | 12.01 |
| Cascade Generalization | 80.8 | 83.1 | 86.8 | 90.5 | 27.0 | 22.0 | 72.89 | 40.43 |
| Voting | 83.2 | 81.1 | 77.3 | 76.2 | 16.7 | 18.41 | 11.52 | 3.12 |
| Grading | 77.4 | 79.5 | 77.4 | 79.8 | 22.65 | 20.2 | 195.59 | 51.11 |
| Stacking | 81.4 | 80.9 | 80.8 | 87.3 | 30.1 | 25.1 | 125.82 | 36.8 |
| BoW | | | | | | | | |
| Ensemble Selection | 78.9 | 66.1 | 87.9 | 74.0 | 31.7 | 40.4 | 77.65 | 17.75 |
| Cascade Generalization | 86.1 | 56.1 | 92.0 | 62.2 | 21.94 | 44.3 | 94.39 | 37.3 |
| Voting | 86.7 | 62.4 | 85.8 | 62.2 | 13.3 | 37.3 | 14.87 | 6.14 |
| Grading | 83.4 | 59.1 | 83.4 | 59.1 | 16.5 | 39.9 | 346.02 | 64.86 |
| Stacking | 86.6 | 55.9 | 85.8 | 63.9 | 23.6 | 56.3 | 205.3 | 50.3 |
| IG | | | | | | | | |
| Ensemble Selection | 80.6 | 66.7 | 88.4 | 74.9 | 31.0 | 40.1 | 47.18 | 32.88 |
| Cascade Generalization | 87.2 | 61.0 | 92.7 | 67.9 | 21 | 43.3 | 146.19 | 108.55 |
| Voting | 89.3 | 64.2 | 84.1 | 61.9 | 10.67 | 35.4 | 24.63 | 18.38 |
| Grading | 85.8 | 62.2 | 85.8 | 60.8 | 14.1 | 39.3 | 262.4 | 101.1 |
| Stacking | 87.1 | 54.5 | 92.5 | 60.6 | 19.47 | 46.9 | 232.62 | 61.05 |

Voting to build a model on both corpora is using tri-gram. It shows the efficiency of the tri-gram method over IG and BoW to build a model.

For the MAE metric, again, the Voting model achieves the minimum values than others on both corpora, which shows that magnitude of the error is significantly less in predicting the labels of both types of articles. The average magnitude error of the Voting model is 18.41% on tri-gram and 10.7% on IG for BET and UFN, respectively. It means that IG is a good feature selection method over other methods on large size UFN corpus while tri-gram is good for small size BET corpus.

To estimate an ensemble model's performance and decide whether a model's performance is acceptable or not, we use a performance ranking metric AUC. On the BET corpus, only $AUC \geq 90$ is achieved by the Ensemble Selection model over the tri-gram feature method. With IG and BoW features the AUC scores of all the other models are $AUC < 75$ which means the performance of these models is acceptable. On UNF corpus, Cascade Generalization achieves outstanding performance to detect fake news with BoW and IG (see Table 4). It achieves 92.0% and 92.7% AUC scores ($AUC \geq 90$) for BoW and IG methods. Cascade Generalization with 86.8% AUC score categorizes its performance ranking to excellent. Again, Ensemble Selection achieves the best AUC score using IG on UFN while Cascade Generalization achieves the best AUC using tri-gram features on BET.

As we are interested to know the performance of a model to predict both labels ("fake", "legitimate") correctly in the corpus, we use balanced accuracy. The maximum BA achieved by a model means that the model is performing more accurately than others to

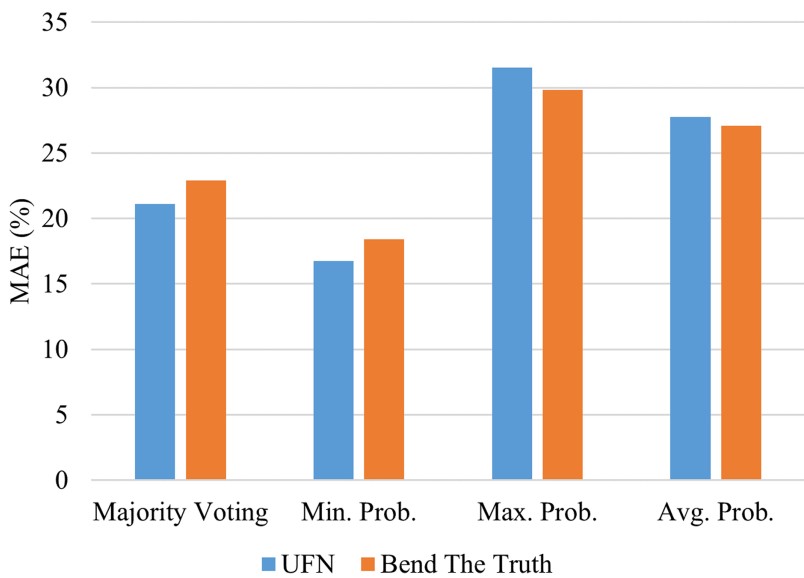

**Figure 3 Performance of Voting model using numerous ensemble rules.**

distinguish fake articles from legitimate articles. Experiment results reveal that Ensemble Selection and Voting outperform the other models on BET corpus and UFN corpora. Ensemble Selection achieves the maximum 83.3% BA on BET corpus using the tri-gram feature On the UFN corpus, the Voting model significantly outperforms the other four ensemble models and achieves an 89.3% BA score using the IG feature. Again, it is noticed that IG outperforms the other methods on UFN while tri-gram outperforms the other feature selection methods on the BET corpus.

Voting model ensemble the numerous base-predictors using some ensemble rule. The four popular rules are majority voting, the product of probabilities, minimum probabilities, and maximum probabilities. By impressive Voting performance on both corpora using balanced accuracy, MAE, and Time metrics, as given in Table 6 and discussed above, we further investigate its performance using different ensemble rules. The mean absolute error values achieved by each ensemble rule is shown in Fig. 3. We conclude that the minimum probabilities rule is impressive to ensemble the predictions of the base-predictors at it achieves the lowest error values on both corpora. The vote model achieves 16.74% and 18.41% MAE scores on UFN and BET corpora. Hence, in our experiments, minimum probabilities, and product of probabilities, both rules perform the same on both corpora.

## Performance comparison of machine learning and ensemble learning models

It is important to know the difference in the performance of ensemble models and individual machine learning models.

A summary of the results achieved by the best ML and EL model with the best feature selection method is given in Table 7. Comparative analysis of the results shows that

**Table 7 Summary of the overall results obtained from the best ML model, EL model, and features.**

|      | Classifier | UFN | Feature | Classifier | BET | Feature |
|------|-----------|-----|---------|-----------|-----|---------|
| Time | SVM | 0.15 | BoW | SVM | 0.17 | IG |
|      | Voting | 11.52 | Tri-gram | Voting | 3.12 | Tri-gram |
| MAE  | SVM | 12.2 | IG | SVM | 23.5 | Tri-gram |
|      | Voting | 10.67 | IG | Voting | 18.41 | Tri-gram |
| AUC  | SVM | 87.3 | IG | SVM | 76.3 | Tri-gram |
|      | Cascade Learning | 92.7 | IG | Ensemble Learning | 91.0 | Tri-gram |
| BA   | SVM | 87.3 | IG | SVM | 76.3 | Tri-gram |
|      | Voting | 89.3 | IG | Voting | 83.3 | Tri-gram |

machine learning models are efficient than EL models and take less time to build a model on both corpora. SVM takes a minimum time of less than a second to build the model on both corpora. Among the EL models, Voting is efficient and takes 11.52 and 3.12 s on UFN and BET. But Voting is much costly than SVM. It is because of the multiple base-predictors in the EL model. EL model combines the three heterogeneous ML models and then the final Voting model predicts the final label based on the prediction of the base-models.

For error analysis, MAE values show that EL models have the lowest values of MAE than individual ML models. Again SVM outperforms the NB and DT by achieving minimum MAE scores on both corpora. On the other side, Voting outperforms the other EL models on both corpora. The lowest score of MAE for EL models means that these models are more accurate in fake news detection. EL models reduce MAE at two levels: at the base-predictors level and ensemble-level. Voting takes the advantage of MAEs of its base-predictor. It reduces the MAE scores of its three base-predictors using minimum probability to predict the final class.

Support Vector Machine achieves maximum scores of AUC 87.3% and 76.3% on UFN and BET. AUC scores rank SVM predictions to excellent on UFN and acceptable on BET. Cascade Learning and Ensemble Learning achieve 92.7% and 91.0% AUC scores on UFN and BET. It categorizes the detection performance of both models as outstanding. SVM outperforms the other ML models and it achieves the maximum BA scores. SVM achieves 87.3% BA on UFN and 76.3% on BET. From EL models, Voting achieves 89.3% BA and outperforms the other EL and ML models on the UFN corpus. On BET corpus, Ensemble Selection models produce 83.3% BA that is the maximum BA among all the models.

The comparison of EL and ML methods using three feature selection methods is interesting valuable. SVM shows the best performance among the three ML models on small and large corpora. SVM achieves the best scores in all the performance measures. Character tri-gram works well on small size corpus BET while IG works well on large size corpus UFN to boost SVM performance. Voting performance is the best performance among EL models using Time and MAE performance measures on both corpora. Ensemble Selection is good at small corpus BET on two performance measures. IG feature

works well with Voting to predict the class of a news article on UFN while tri-gram is the best with Voting and Ensemble Learning. Further, it can be seen that the IG feature works well on large size corpus while character tri-gram is good on small size corpus.

## CONCLUSIONS

Fake news detection through ensemble models is the most prominent topic of machine learning. If the traditional machine learning models are used for fake news detection task, the performance is not encouraging because their performance is limited to corpus characteristics. In this study, we deliberately choose ensemble methods to classify fake and legitimate news articles of the Urdu language. First, we use three machine learning models to classify two fake news corpora. Our experiments on two Urdu news corpora conclude that the individual machine learning model SVM outperforms the DT and NB on both corpora. SVM achieves the best scores of balanced accuracy and AUC and the minimum score of MAE. Second, we use five ensemble models for the same task.

We find that ensemble models with three base-predictors DT, NB, and SVM, Ensemble Selection, and Vote models outperform the other on BET and UFN corpora, respectively. After the analysis of MAE, AUC, time, and BA values, we conclude that Voting with minimum probability is the best EL model for the fake news detection task. IG feature works well with large size corpus while character tri-gram works well on small size corpora.

This study has several limitations that need to be addressed in future studies. The proposed corpus UFN still needs to grow by adding more news articles to it. We used online Google translation API in English-to-Urdu translation and we believe that translation accuracy is not a hundred percent. A study is vital in the future to explore the translation accuracy and quality of various translation APIs like Baidu, Google, etc. The potential of deep learning models also can be explored to detect fake news for Urdu. Further, we also hope to design another multilingual fake news corpus of English and Urdu news articles.

### Funding

This work was supported by the Research and Development Plan of Shaanxi Province under Grant 2017ZDXM-GY-094 and Grant 2015KTZDGY04-01, and in part by the National Natural Science Foundation of China under Grant 61972321. There was no additional external funding received for this study. The funders had no role in study design, data collection and analysis, decision to publish, or preparation of the manuscript.

### Grant Disclosures

The following grant information was disclosed by the authors:
Research and Development Plan of Shaanxi Province: 2017ZDXM-GY-094 and 2015KTZDGY04-01.
National Natural Science Foundation of China: 61972321.

## Competing Interests

The authors declare that they have no competing interests.

## Author Contributions

- Muhammad Pervez Akhter performed the experiments, performed the computation work, prepared figures and/or tables, and approved the final draft.
- Jiangbin Zheng conceived and designed the experiments, performed the computation work, prepared figures and/or tables, and approved the final draft.
- Farkhanda Afzal conceived and designed the experiments, analyzed the data, authored or reviewed drafts of the paper, and approved the final draft.
- Hui Lin conceived and designed the experiments, performed the experiments, analyzed the data, performed the computation work, authored or reviewed drafts of the paper, and approved the final draft.
- Saleem Riaz analyzed the data, performed the computation work, prepared figures and/or tables, authored or reviewed drafts of the paper, and approved the final draft.
- Atif Mehmood performed the experiments, prepared figures and/or tables, and approved the final draft.

## Data Availability

Data is available in the Supplemental Files.

The generated machine learning, ensemble models, and data are available at GitHub: https://github.com/pervezbcs/Urdu-Fake-News.

A well-known data mining tool, WEKA (https://www.cs.waikato.ac.nz/~ml/weka/) was used for these experiments.

## Supplemental Information

Supplemental information for this article can be found online at http://dx.doi.org/10.7717/peerj-cs.425#supplemental-information.

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
