# Peer review of "Supervised ensemble learning methods towards automatically filtering Urdu fake news within social media"

_PeerJ Computer Science, doi:10.7717/peerj-cs.425_

## Round 0.1 · original submission · Major Revisions

Please carefully revise your manuscript according to the major comments of the reviewers. In particular, more results or explanations are required to (1) address the accuracy of the English-to-Urdu translation and (2) to validate the effectiveness of ensemble learning methods compared to the single classifiers.

Reviewer 1 ·

Basic reporting

This paper utilizes ensemble methods to classify fake and legitimate news articles of the Urdu language. Authors built an annotated news corpus composed of Urdu news articles using Google Translate. Moreover, they perform experiments with traditional machine learning classifiers and five ensemble models to detect fake news and comparatively analyze the performance of their models. The language used in this paper is clear and unambiguous. Their work, innovations, contributions, and results are clearly presented in the introduction section, and the whole structure meets the requirements. However, the resolution of the figures needs to be improved, especially Figure1&2. In addition, all of the captions for the figures and tables are written as 'Figure 1' and 'Table 1'.

Experimental design

This paper presents a meaningful research topic and fills the gap in the current knowledge gap with high relevance for real-life situations. This paper innovatively builds a corpus using translated labelled English corpus, which addresses the problem that the scarcity of Urdu language corpus. However, there exist two problems: first, the accuracy of the English-to-Urdu translation is not verified. As mentioned in lines 461-462, the idea that all models perform better on UFN than BET is not a sufficient reason to illustrate the accuracy of Google translate API. Second, line 351 mentions that they randomly selected news articles. Why they were randomly selected instead of using all the data?

Validity of the findings

All underlying data have been provided and the conclusions are well presented. The final experiments are able to demonstrate that they design a classifier that performed well on their dataset.

Additional comments

This paper presents a meaningful research topic and fills the gap in the current knowledge gap. Authors built an annotated news corpus composed of Urdu news articles using Google Translate, perform experiments with traditional machine learning classifiers and ensemble models to detect fake news. However, the section about corpus building needs to be supplemented to illustrate the accuracy of the methodology. In addition, there are some typos:
line 38: is a fake news: a piece of fake news.
line 441: accruacy: accuracy
line 499: shown: shown in.
Moreover, the resolution of the figures needs to be improved and all captions of the tables and figures need to be revised.

·

Basic reporting

In general, this paper is easy-to-follow. However, one concern is on the use of "Feature Selection"; I think the authors meant extracting the features (feature extraction) rather than feature selection (for dimensionality reduction).

It has a comparatively comprehensive reference list, while more advanced relevant work with the topic of fake news detection could be considered, e.g., neural-network-based studies.

Most figures and tables are clear. It would be more professional if (1) accurately numbering figures and (2) just like other figures, note "%" in Figure 3.

I didn't find access to the raw data.

Experimental design

This work is within the Aims and Scope of the journal. It contains a detailed experimental setup and some of the necessary experiments. However, the experiments are insufficient. If the authors aim to validate the effectiveness of ensemble learning methods compared to the single classifiers. Multiple groups of features are requested for validation. For example, separating the combined n-grams or trying other common feature sets, like LIWC.

Validity of the findings

The work's limitations have not been discussed yet. For example, the Urdu news articles were obtained not directly but by translating the English news articles. The translation quality is hard to be perfect; the data might need to be further improved from this perspective.

Additional comments

In general, this work contains most of the necessary information, components, and analyses. However, my major concern is about its innovation since rather than proposing news methods, the authors mainly use some existing methods to detect fake news. I suggest the authors explicitly claim their innovation in the paper.

---

## Round 0.2 · Minor Revisions

There are a few minor comments from the reviewers which need to be addressed before the manuscript can be accepted. I hope that you can quickly deal with these comments and return your manuscript to us.

Reviewer 1 ·

Basic reporting

In general, this version of the paper has been improved both in terms of content and writing. Several parts have been rewritten and now the quality of the paper is higher than before.

Experimental design

The authors address my comments satisfactorily, both the accuracy of the English-to-Urdu translation and the news articles selection.

Validity of the findings

The authors have added the limitations of this study to the manuscript, which makes it more reasonable.

Additional comments

The authors did some progress with the new changes, and I believe that it would be accepted.

·

Basic reporting

Please see my general comments for the author.

Experimental design

No comment.

Validity of the findings

No comment.

Additional comments

I appreciate the authors have addressed most of my previous concerns. Nevertheless,

1. the writing can be further improved before being accepted;

2. the review for low-resource language datasets is insufficient, where several newly released ones for fake news research are suggested to include, such as CHECKED and MM-COVID. Meanwhile, I hope the following surveys could help the authors with the review:
- Zhou, Xinyi, and Reza Zafarani. "A survey of fake news: Fundamental theories, detection methods, and opportunities." ACM Computing Surveys (CSUR) 53.5 (2020): 1-40.
- Sharma, Karishma, et al. "Combating fake news: A survey on identification and mitigation techniques." ACM Transactions on Intelligent Systems and Technology (TIST) 10.3 (2019): 1-42.

---

## Round 0.3 · accepted · Accept

There are no further comments on your manuscript.